# Biomarkers for Homologous Recombination Deficiency in Cancer

**DOI:** 10.3390/jpm11070612

**Published:** 2021-06-28

**Authors:** Svenja Wagener-Ryczek, Sabine Merkelbach-Bruse, Janna Siemanowski

**Affiliations:** Institute of Pathology, University Hospital Cologne, D-50924 Cologne, Germany; sabine.merkelbach-bruse@uk-koeln.de

**Keywords:** DNA double-strand break, BRCA, homologous recombination, HRD score, BRCAness, PARPi

## Abstract

DNA double-strand breaks foster tumorigenesis and cell death. Two distinct mechanisms can be activated by the cell for DNA repair: the accurate mechanism of homologous recombination repair or the error-prone non-homologous end joining. Homologous Recombination Deficiency (HRD) is associated with sensitivity towards PARP inhibitors (PARPi) and its determination is used as a biomarker for therapy decision making. Nevertheless, the biology of HRD is rather complex and the application, as well as the benefit of the different HRD biomarker assays, is controversial. Acquiring knowledge of the underlying molecular mechanisms is the main prerequisite for integration of new biomarker tests. This study presents an overview of the major DNA repair mechanisms and defines the concepts of HRR, HRD and BRCAness. Moreover, currently available biomarker assays are described and discussed with respect to their application for routine clinical diagnostics. Since patient stratification for efficient PARP inhibitor therapy requires determination of the *BRCA* mutation status and genomic instability, both should be established comprehensively. For this purpose, a broad spectrum of distinct assays to determine such combined HRD scores is already available. Nevertheless, all tests require careful validation using clinical samples to meet the criteria for their establishment in clinical testing.

## 1. Introduction

The development of cancer is a multifactorial process and comprises key events such as the ability of unlimited replication, the induction of angiogenesis and the activation of invasion and metastasis. Different malignant features are progressively accumulated [1]. One major underlying mechanism is the emergence of genomic instability caused by genetic mutations arising from exogenously or endogenously caused DNA damage or failures in DNA damage repair. Healthy cells maintain genomic integrity by a variety of repair mechanisms, each addressing unique forms of DNA damage. Incorrect bases incorporated during replication are removed by proteins of the mismatch repair (MMR) pathway. The nucleotide excision repair (NER) mechanism identifies structural distortions within the DNA double-strand and removes the affected bases. The base excision repair (BER) pathway is activated by damaged DNA bases. In response to double-strand breaks, two different repair pathways are available; the exact mechanism of homologous recombination repair (HRR) and the error-prone non-homologous end joining (NHEJ) [2].

Defects in the DNA repair system are an underlying cause of genomic instability due to the accumulation of genetic changes [3]. Homologous recombination repair deficiency (HRD) resulting in DNA double-strand breaks is considered to be the most lethal of all DNA repair defects since cancer cells switch to the error-prone NHEJ pathway fostering genomic instability and cell death. Therefore, HRD is therapeutically addressed by poly (ADP-ribose) polymerase (PARP) inhibition. PARP inhibition (PARPi) disables single-strand break repair and leads to further accumulation of double-strand breaks, consequently requiring homologous recombination repair. Positive outcome results in clinical trials led to the approval of several inhibitors for different tumor entities. Nevertheless, it remains challenging to define those patients who might benefit from PARPi therapy. In particular, a better understanding of the underlying mechanisms leading to HRD and the identification of patients beyond the *BRCA1* and *BRCA2* mutated cohort with the so-called BRCAness phenotype are crucial points to be addressed.

This review article provides an overview of the fundamental mechanisms of DNA repair. Important terms such as HRR, HRD and BRCAness are defined and analysis methods are described, especially with regard to their integration in molecular pathology routine diagnostics.

## 2. DNA Repair Mechanisms

Mammalian cells obey different cellular responses to DNA damage, including DNA damage repair, cell cycle arrest and apoptosis. Regarding the responses of DNA repair, different mechanisms can take place. Different types of DNA damage trigger different, specific repair mechanisms (Figure 1). The simplest way of DNA repair is proofreading or direct reversal repair, occurring during replication. Hereby, polymerases are able to proofread their synthesized nucleotide sequence with their exonuclease activity (direct repair in 3′–5′direction). For example, primary alkylation damage reversal involves proofreading. The direct reversal of such lesions is mediated by the methylguanine methyltransferase (MGMT). The methyl group is transferred to a catalytically active cysteine and afterwards MGMT is targeted to the proteasome for degradation [4].Nevertheless, it requires more sophisticated repair mechanisms to effectively correct complex damage on already replicated DNA caused by several endogenous or exogenous factors.

### 2.1. Single-Strand Lesion Repair

Depending on the factor that causes DNA damage, different types of mismatches/errors occur on single-stranded DNA. For example, radiation modifies single bases or DNA adducts alter the conformation of the DNA. Those alterations are detected by proteins associated with the nucleotide excision repair mechanism (NER). This mechanism allows the excision of the damaged DNA segment with subsequent synthesis of a new strand and followed by ligation to successfully reconstitute an intact DNA double-strand. Mismatches created during the replication process due to slippage of DNA strands are corrected by the so-called mismatch repair. Mismatched bases or reading frame shifts are recognised by specific protein dimers, consisting of MSH2 and MSH3 or MSH6. Another protein dimer of MLH1 and PMS2 facilitates cutting of the strand in cooperation with further enzymes such as helicases and endonucleases. The resulting gap is filled with the matching nucleotides by polymerases and the ligation of ends finalizes this process.

DNA damage of single bases caused by oxidation, deamination or alkylation are commonly repaired by base excision repair (BER). In a first step, the damaged base is recognised and removed by specialized DNA glycosylases. This step results in an (AP) site that is cleaved by AP endonucleases to yield a DNA nick (3´hydroxyl adjacent to a 5´deoxyribosephosphate). At this point, the enzyme called Poly (ADP-ribose) polymerase (PARP) binds to the gapped DNA strand and engages proteins required for strand protection and repair. Filling of the gap is mainly facilitated by DNA polymerase ß (Polß). Finally, after dissociation of PARP and excision, the correct base can be attached to the strand by a polymerase. Following this, the DNA strand is sealed by ligases in complex with X-ray repair cross-completing protein 1 (XRCC1) in the short-patch BER. Short-patch BER is a prerequisite for single nucleotide replacement, whereas long-patch BER utilizes a displacing synthesis of 2–10 new nucleotides [5,6,7,8].

### 2.2. Double-Strand Break Repair

DNA double-strand breaks (DSB) can be caused by either internal processes, ionizing radiation or chemotherapeutics. This type of damage can lead to extensive chromosomal changes and apoptosis. Two major repair mechanisms have evolved to deal with double-strand lesions, homologous recombination repair (HRR) and non-homologous end joining (NHEJ). The nature of DSBs caused by internal factors, such as replication block, and external influences, such as ionizing radiation, is obviously different. On one hand, lesions associated with the replication process can be directly repaired by using the sister chromatid, which is located in close proximity, as a template for homologous recombination (HR). On the other hand, DNA damage caused by ionizing radiation mainly occurs at the densely packed chromatin structure. Therefore, an interaction with intact homologous sequences as a template for HR is not possible. In order to deal with such types of lesions on DNA in condensed chromatin structures, vertebrates frequently use NHEJ to simply re-ligate the DSB end strands. Another likely reason for NHEJ being the predominant DSB repair mechanism in humans might be the following fact: HR, as mentioned above, requires the sister chromatid as a repair template present following DNA replication. Therefore, HR preferably takes place in the S/G2 phase. In contrast, NHEJ is dominant in the G1 phase, which is longer than S/G2.

#### 2.2.1. Homologous Recombination Repair

Homologous recombination repair (HRR) is an accurate repair mechanism to cope with DSBs. In a first step, the lesion is recognised by the MRE11-RAD50-NBS1 (MRN) complex, which activates ATM (Ataxia telangiectasia mutated) kinase. Upon DNA 5′-end resection, the replication protein (RPA) coats the single-strand DNA regions and activates ATR (Ataxia telangiectasia and Rad3-related) kinase. In the following, RPA is replaced by RAD51 with the help of further repair-associated proteins, such as CHEK2, BRCA1, BRCA2 and PALB2, which are loaded with RAD51. Subsequently, the defective DNA strand attaches to its sister chromatid, which is used as a template for DNA resynthesis. This mechanism of DSB repair is restricted to late S- and G2 phases of the cell cycle [9,10].

#### 2.2.2. Non-Homologous End Joining

The second pathway that evolved to cope with DSBs belongs to the simplest mechanisms: non-homologous end joining (NHEJ). This mechanism is commonly applied by the cells during the G1 phase. The core component of this pathway is the Ku70–Ku80 complex, which is able to bind ends of double-strand broken DNA and recruit DNA-PKcs to initiate NHEJ. Further factors process the broken DNA to build DNA ends compatible for ligation. In a final step, DNA ligase IV (LIG4) is simultaneously recruited with XRCC4 and an XRCC4-like factor (XLF) to ligate the processed DNA and restore genome integrity. Nevertheless, loss of genetic information at the break is highly frequent. For example, changes within the DNA sequence or ligation of random blunt ends may take place. Thereby, chromosomal integrity is prone to get lost, giving rise to chromosomal rearrangements [2,11].

### 2.3. Deficiency of Homologous Recombination Repair and BRCAness

The impairment of HRR activity is called homologous recombination repair deficiency (HRD) and is caused by different factors. Genes involved in the HRR pathway, such as *BRCA1/2* or *PALB2,* to repair DSB can be either mutated within the tumor (somatically) or germline, thus causing HRR impairment. Additionally, expression of the HRR associated proteins can be diminished by promoter methylation, for example [12]. In order to deal with this impairment, cells tend to activate the alternative NHEJ pathway for DSB repair. As already mentioned, this pathway is prone to ending in chromosomal rearrangements. The resulting genomic instability can be observed as genomic scars [13].

The most common pathogenic alterations of HRR associated genes occur within the genes *BRCA1* and *BRCA2* [14]. Nevertheless, other genes of the pathway and their associated proteins can be affected as well. This phenotype of homologous recombination repair deficiency independent of *BRCA1/2* mutations is referred to as BRCAness [10].

## 3. PARP Inhibition and Homologous Recombination Repair Deficiency

### 3.1. The Concept of Synthetic Lethality

Utilizing poly (ADP-ribose) polymerase (PARP) inhibitors in BRCA deficient cells has shown great promise in clinical studies for patients with *BRCA1/2* mutated tumors. Meanwhile, several PARP inhibitors have been approved for the treatment of ovarian, breast, prostate and pancreatic cancers in different clinical settings. With the recent approval of the combination of olaparib and bevacizumab as a maintenance treatment for high-grade epithelial ovarian cancer based on the data of the PAOLA-1 trial, HRD positivity was added as a new biomarker for patient stratification [15]. For metastatic castration-resistant prostate cancer (mCRPC), olaparib received FDA approval in patients with either mutations in *BRCA1*, *BRCA2*, *ATM* or one out of 12 other HR-related genes. This indication was based on the results gathered from the PROfound trial [16].

PARP1 and PARP2 add long chains of poly (ADP-ribose) to several proteins and PARP itself, a process which is called PARylation. PARP inhibitors prevent the modification of PARP proteins, thereby impeding dissociation of PARP from the DNA single-strand break. Furthermore, attachment of additional repair proteins is inhibited, leading to accumulation of single-strand breaks. In replicating cells these single-strand breaks are converted into double-strand breaks [17]. Besides the inhibition of single-strand repair, PARP inhibitors bind PARP1 and PARP2 enzymes to damaged DNA. This so-called PARP trapping holds a strong cytotoxic effect. PARP-DNA complexes result in a stalled replication fork and further DNA double-strand breaks accumulate. In healthy cells these double-strand breaks are eliminated by the mechanism of homologous recombination repair (HRR) [18].

Cells with HRD caused by alterations in the genes encoding HR proteins are forced to use the error-prone pathway of non-homologous end joining (NHEJ). This leads to the accumulation of genomic damage, thus PARP inhibition is particularly effective in the absence of an intact HR pathway [2,19]. DNA double-strand breaks associated with PARP inhibition and replication lead to chromosomal rearrangement, genomic instability and apoptotic cell death. The concept of combining two conditions to force cell death is known as synthetic lethality and its use in tumors with defective DNA repair or altered checkpoint controls was described by Ashworth in 2008 for the first time [20]. Initially, only mutations in *BRCA1* and *BRCA2* were introduced as predictive biomarkers for successful PARP inhibition. Interestingly, in a significant proportion of tumors holding HRD, variants within these two genes could not be detected [15]. Therefore, analysis of genomic instability as a read-out for alterations in the HR pathway was additionally proven as a suitable biomarker for a response to PARP inhibition.

### 3.2. Biomarkers for PARPi: Genes Involved in the Homologous Recombination Repair Pathway

The most common alterations currently known to confer sensitivity to PARP inhibition are loss of function mutations in *BRCA1* and *BRCA2* [21]. BRCA1 can also be inhibited by gene promoter methylation but, regarding responses to PARP inhibitor therapy, retrospective and preclinical data showed conflicting results. In ovarian cancer patients, *BRCA1* methylation was not associated with long-term responses [22], whereas, in patient-derived xenografts, a homozygous *BRCA1* methylation status led to PARP inhibitor sensitivity [23].

Additionally, preclinical data suggested that patients whose tumors show HRD caused by other mutations in the HR pathway, may benefit from therapy with PARP inhibitors [18]. Several different types of proteins are involved in protecting genome integrity and present possible biomarkers for PARP inhibition. Kinases, such as ATM or ATR, are responsible for damage recognition; a second group are signal mediators, such as CHEK2 and BRCA1; finally, repair is initiated by effector proteins, such as BRCA2 and RAD51. Other proteins, such as PALB2 and BRIP1, function as facilitators of the HR pathway [15].

The prevalence of mutations in HR genes, apart from *BRCA1* and *BRCA2,* in all solid tumors was largely unknown and differed between published data. Therefore, Heeke et al. (2018) carried out a comprehensive molecular profiling in a large cohort of more than 17,000 solid tumor specimens [14]. Their study revealed pathogenic mutations in HR pathway genes in 17.4% of tumors with endometrial cancer being most commonly mutated (34.4%), followed by biliary tract (28.9%), bladder (23.9%), hepatocellular (20.9%) and ovarian cancer (20.0%). The most frequently mutated HR genes included *ARID1A* (7.2%), *BRCA2* (3.0%), *BRCA1* (2.8%), *ATM* (1.3%), *ATRX* (1.3%) and *CHEK2* (1.3%). Several of these mutations correlate with clinical responses to PARP inhibitor treatment. Recently, it was shown that patients with *PALB2* mutated metastatic breast cancer benefit from PARP inhibition in contrast to patients with exclusive *ATM* or *CHEK2* mutations [24]. Seventeen percent to twenty-five percent of pancreatic cancers harbor mutations in genes related to DNA repair [25] and recent clinical trials suggested the benefit of PARP inhibitor treatment for patients with platinum-sensitive pancreatic cancer genomic alterations in DNA repair genes beyond *BRCA* [26].

Mutational analysis of *BRCA1*, *BRCA2* and additional HR-related genes can be performed on either blood for germline testing or tissue samples for both germline and somatic testing. The decision, whether germline or somatic mutation status, has to be determined depending on the specific approval of a PARP inhibitor for a certain tumor entity [27]. Tissue testing is routinely carried out on formalin-fixed paraffin-embedded tissue (FFPE), thus the extracted DNA may be strongly degraded and insufficient for the detection of large gene deletions. Moreover, FFPE tissue testing is hampered by the occurrence of fixation artifacts, which may lead to false-positive results. On the other hand, blood testing only detects germline mutations and is not suitable for simultaneous analysis of the HRD phenotype.

The above described alterations can be detected by parallel sequencing with targeted gene panels. Different pre-assembled panels are commercially available from different providers (Table 1), but a custom design is also possible. Whereas the detection of these mutations seems to be technically manageable, interpretation of mutations in the HR-related genes in a clinical context remains challenging. For *BRCA* variants, the internationally agreed classification scheme of the International Agency for Research on Cancer (IARC) [28], the regulations of the ENIGMA consortium (Evidence-based Network for the Interpretation of Germline Mutant Alleles) and databases, such as BRCA Exchange [29], aid in variant classification. For non-*BRCA* variants, the same classification scheme can be used [30], but databases are currently being assembled and still include a high number of variants of unknown significance.

Interpreting the consequences of alterations in HR-related genes in different tumor entities remains challenging and clinical data show that mutations in specific genes can be of predictive value for only certain entities [31]. The underlying DNA repair mechanisms are complex, thus the role of a single protein as a predictive biomarker depends largely on its specific function within the repair process. Additionally, the low frequency of the individual mutations in different tumor entities complicates the assessment of their relevance. Considering the current state of knowledge testing for alterations in HR-related genes should always include a bunch of those genes and single gene approaches should be avoided.

The assessment of the predictive value of mutations in HR-related genes beyond *BRCA1* and *BRCA2* depends on the evaluation of further prospectively collected data. Whereas the FDA has already approved the use of PARP inhibitor therapy for mCRPC with mutations in HR-related genes based on the data of the PROfound trial [16], the respective ESMO working group stated in their recommendations that the evidence for clinical validity of non-*BRCA* HRR genes is currently too low [21].

### 3.3. Biomarker for PARPi: Genomic Instability

#### LOH- TAI- LST

*BRCA1* and *BRCA2* mutations are commonly used to identify patients with HRD. In a recent clinical trial in ovarian cancer it was shown that nearly 20% of the study population was HRD positive without having a *BRCA* mutation [15]. As these patients also benefit from PARP inhibition, assays identifying HRD without knowing the underlying mechanisms should be applied. HRD causes patterns of mutations and deletions/insertions (mutational signatures) as well as copy number variations (CNV) and structural rearrangements, effects which can be analyzed by different molecular methods. The larger genome alterations are called “genomic scars“ [32] and are currently the basis of available clinical assays for HRD, which can only be performed on tumor tissue. These genomic scars comprise loss of heterozygosity (LOH), telomeric allelic imbalance (TAI) and large scale state transitions (LST). LOH occurs by either deletion of one allele (copy loss LOH) or by deletion and simultaneous duplication of the remaining allele (copy neutral LOH), resulting in the loss of one of the two alleles at a heterozygous locus. TAI, telomeric allelic imbalance, is defined as the number of regions with allelic imbalance extending to the subtelomere but not crossing the centromere. (Figure 2) [33,34,35]. HRD-related genomic alterations have to be distinguished from other genomic alterations found in cancer genomes. Thus, measures for these three markers in the context of HR-deficient tumors have been defined using single nucleotide polymorphism (SNP) arrays in the respective cohorts (Figure 2). Abkevich and colleagues (Br. J. Cancer, 2012) showed that the number of subchromosomal segments with LOH of a size exceeding 15 Mb but shorter than the length of a whole chromosome is associated with a functional inactivation of BRCA1, BRCA2 or RAD51C [33]. Telomeric allelic imbalances extend from the double-strand breakpoint to the subtelomeric region of a chromosome without including the centromere. High levels of such aberrations are shown in tumors with BRCA1 and BRCA2 deficiency as well as tumors sensitive to platinum chemotherapy [34]. LST is defined as a third signature by Popova et al., (Cancer Res. 2012) as the number of break points occurring between adjacent regions of at least 10 Mb. This marker was established in breast tumors and cell lines with BRCA deficiency [35].

LOH, TAI and LST were found to be independently associated with homologous repair deficiency, but the combination of all three scores allows for the most robust prediction [36]. A composite HRD score was developed in three TNBC (triple negative breast cancer) clinical trial cohorts using the unweighted sum of the three single scores. Furthermore, a threshold for HRD positivity was selected based on the likelihood of response to platinum-based chemotherapy [37].

A major limitation of the genomic scar assays is the potential lack of representing the current HRD status when analyzing archival tumor tissue. Tumor cells previously defined to be HR-deficient might have restored their proficiency by resistance mechanisms, such as reversion mutations in HR-related genes. RAD51 focus formation or replication fork assays could be used as functional HRD assays. Nevertheless, there are still some challenges to be solved prior to implementing such assays into routine diagnostics, such as the need for fresh tissue and standardization of positivity thresholds [38].

Commercially available tests most often combine *BRCA* mutation testing with either a composite HRD score or the assessment of LOH, as described above (Table 1). To date, only the Myriad myChoice assay and the Foundation Focus CDx BRCA LOH assay were used for patient stratification in clinical studies [39]. Thus, alternative assays must be validated with clinical samples previously analyzed with one of the above-mentioned assays.

## 4. Methods for Detecting DNA Repair Defects as Biomarkers for PARP Inhibition

As described above, chromosomal instability can be due to changes in the HR-related genes including *BRCA1*, *BRCA2, ATM, BARD1*, *BRIP*, *CDK1*, *PALB*, *CHEK1*, *CHEK2*, *FANCL*, *PPP2R2A*, *RAD51B*, *RAD51C*, *RAD51D* and *RAD54L*. Another possibility for revealing damage of the genome is the estimation of an HRD score. While alterations of the HR-related genes are evaluated by standard methods of mutation or expression analyses, the HRD score can be computed by combining the measurements of larger genomic defects including LOH, TAI, and LST [37].

In order to detect DNA repair defects individually or concomitantly, a broad range of different technologies is nowadays available.

### 4.1. Single Nucleotide Polymorphism (SNP) Arrays

SNP arrays are one of the most primary assays capable of measuring chromosomal abnormalities, such as CNVs, as well as the above-mentioned genomic scars LOH, LST and TAI. Their major disadvantage is their lack of reliable *BRCA1* and *BRCA2* mutation detection.

The chromosomal microarray analysis includes an SNP-based microarray where SNP probes are immobilized on a special matrix followed by hybridization. Differences in fluorescent signals can be detected by an array reader [40].

Those genome-wide polymorphism detection assays enable simultaneous analysis of up to 850,000 single nucleotide polymorphisms [41]. Albeit using different chemistries, two commonly used SNP arrays, including OncoScan Dx (Affymetrix) and Infinium CytoSNP-850K BeadChip v1.2 (Illumina), can be applicated to determine LOH, LST and TAI after bioinformatic analysis, using either 80 ng or 250 ng FFPE DNA, respectively [42,43,44,45,46,47].

### 4.2. Whole Genome Sequencing

Comprehensive HRD testing, including both the detection of alterations of HRR genes and an HRD score, can be conducted by parallel sequencing (PS) tools.

One option that has already revealed good correlation between SNP array-based and PS-based HRD scores, is the unbiased whole genome sequencing (WGS) using mostly 100 ng to 500 ng of paired normal and tumor DNA [48].

WGS facilitates the detection of somatic mutations, rearrangements and CNVs [49]. Besides identification of potentially pathogenic *BRCA* alterations, this testing approach allows the detection of LOH, LST and TAI. In combination with a suitable bioinformatics approach based on evaluation algorithms, such as HRDetect and scarHRD, HRD scores including all three parameters can be estimated [48,50]. With the Classifier of Homologous Recombination Deficiency (CHORD) algorithm, at least LOH status can be determined for possible HRD detection [51].

Although parallel sequencing is a commonly used tool in routine diagnostics, WGS analysis used in this way is only cost effective when using sequencing platforms, such as Illumina’s HiSeq or NovaSeq. Since the number of laboratories capable of such broad and exclusive equipment is still limited, a more cost-effective alternative constitutes WGS at low coverage. This so-called shallow or low-pass WGS is defined by the reduction of a reading depth to 0.1–3×, allowing lower sequencing capacity. Using the shallow HRD algorithm, LOH, LST and TAI can still be detected even though detection of mutations in *BRCA1/2* on a reliable level is no longer possible, caused by the lowered coverage [52]. In spite of down sampling, a good correlation between HRD scores of SNP array, WGS and shallow WGS is approved [53].

### 4.3. Targeted Panel Sequencing

Nowadays, most HR-related genes listed above are integrated in comprehensive, targeted parallel sequencing panels (customized or off-the-shelf) and range in size from 13 to larger than 500 genes (Table 1). In order to detect large deletions and insertions in exonic/intronic boundaries, hybrid capture-based parallel sequencing technologies are preferably used for mutation detection in contrast to amplicon-based ones [54]. The TruSight Tumor170 or TrueSight Oncology 500 from Illumina as well as different Oncomine panels from Thermo Fisher Scientific provide the opportunity of using large HRR gene panels based on different technologies across laboratories, thereby facilitating fast laboratory integration. If an in-house bioinformatic pipeline is not available, then all mentioned panels also offer a bioinformatic solution from the manufacturers’ site.

Since comprehensive cancer panels are still not as common as assumed due to reimbursement difficulties for gene analysis without diagnostic markers in several countries, some manufacturers offer the analysis of *BRCA1/2* mutations exclusively or in combination with LOH status and/or the combined HRD score.

The FoundationFocus CDxBRCA LOH is performed by a centralized laboratory using 50 ng DNA and designed to reveal the *BRCA1/2* mutation status and an HRD-LOH score. Patients with an LOH score ≥ 16 are classified as LOH high and <16 as LOH low. Regarding this threshold, HRD positivity is either defined by the *BRCA* mutation status or LOH high ranking of a sample. An alternative testing system for a local application is the Oncomine Comprehensive Assay Plus from Thermo Fisher, which can also determine both *BRCA* status and an HRD-LOH score [55].

Nevertheless, a combined score of LOH, TAI and LST has a higher clinical impact than the LOH score alone and is therefore recommended [21,36].

In order to define a genomic instability score (GIS) composed of LOH, LST and TAI (HRD score) and to provide the *BRCA1/2* mutational status at the same time, Myriad’s myChoice panel is designated as a precursor test for comprehensive HRD testing [56].

This parallel sequencing-based approach uses around 30–200 ng of a tumor sample for the detection of more than 54,000 SNPs and mutational alterations including nucleotide variants (SNVs), indels and large rearrangements in the *BRCA1* and *BRCA2* genes [36]. A positive HRD status can either be determined by the presence of a pathogenic or likely pathogenic mutation in *BRCA1* or *BRCA2* and/or by a dichotomous calculated GIS of ≥42 [37]. This threshold has already been comprehensively evaluated in several clinical trials [18,57,58].

Since the myChoice panel is only available in central laboratories, different manufacturers are currently developing HRD detection assays, including *BRCA1/2* mutational status assessment and HRD scores comparable to the myChoice panel, in order to allow independent testing in experienced laboratories.

One of those tests already available is the HRD Focus Assay provided by Amoy Dx using 100 ng of FFPE or blood DNA. This assay also allows the simultaneous analysis of SNVs and indels in the whole coding regions and exon–intron boundaries of *BRCA1* and *BRCA2,* and estimates a genomic scar score (GSS) based on the analysis of 24,000 SNPs [59]. With the help of a locally installed analysis software (ANDAS), the user is able to define this score using a threshold of GSS ≥50 as being HRD positive and <50 as being HRD negative if no *BRCA* mutation can be detected. In order to separate somatic from germline mutations, an input of paired normal and tumor tissue is still recommended. Favorably, library preparation has a short turnaround time and sequencing can be performed on common sequencing platforms such as the benchtop sequencer NextSeq (Illumina). Amoy Dx additionally offers an HRR panel comprising 32 different genes of the HRR pathway.

The development of additional assays focusing on HRD analysis with a combined detection system for *BRCA1/2* mutations and LOH, LST and TAI from providers, including Qiagen and Agilent, is in progress.

## 5. Conclusions and Future Perspectives

Tumors without functional homologous recombination repair seem to be more susceptible to PARP inhibitors. PARP inhibition results in the accumulation of unrepaired DNA single-strand breaks that are converted into double-strand breaks during replication. The combination of homologous recombination repair deficiency and PARP inhibition leads to tumor cell death. A reliable and accurate way to measure HRD across all tumor entities is needed but is still missing. Beyond *BRCA1* and *BRCA2* mutations, alterations in other HR-related genes, as well as inactivation by promoter methylation and large genomic defects, so-called genomic scars, are used as biomarkers to stratify patients for PARP inhibitor therapies. Currently, it seems that this large variety of biomarkers and tests used for the assessment remain limited in their ability to effectively identify all patients who might benefit from PARP inhibitor treatment (Figure 3).

Besides centrally performed commercial tests, which were used in clinical trials, alternative assays are entering the market, but significant variation may occur by using different methodologies, such as SNP arrays or different parallel sequencing panels. For the evaluation of whole genome sequencing, complex bioinformatic algorithms were developed, but the resulting scores are often not comparable to scores used within clinical studies. For routine diagnostics, targeted panels might be a better solution. They are either already available as newly designed assays or currently under development as add-ons for existing panels from different providers. Nevertheless, for all alternative assays not yet tested in clinical studies, careful clinical validation is needed to guarantee accuracy and reliability. Modern parallel sequencing techniques allow for simultaneous detection of mutations in HR-related genes, mutational signatures, genomic scars and resistance mechanisms, such as reversion mutations. Thus, the development of composite biomarker assays may improve patient stratification.

Functional tests for assessing current activity of HR repair, such as the RAD51 assay, are under way. They may complement the molecular HRD testing and have the advantage of being faster and cheaper; thus, they might be used for screening purposes in the near future. Another type of functional assay is the DNA fiber assay that can be employed to determine the fork degradation phenotype. Work is in progress to bring these assays into routine diagnostics, but standardization and a prospective comparison with the currently used assay is still needed.

In summary, the optimal patient selection for each treatment remains one key objective of further clinical research. There is a high need for additional or combined predictive biomarkers for the stratification of patients who benefit from PARPi therapy, as well as for broad implementation of HRD Testing and for other tumor entities.

## Figures and Tables

**Figure 1 jpm-11-00612-f001:**
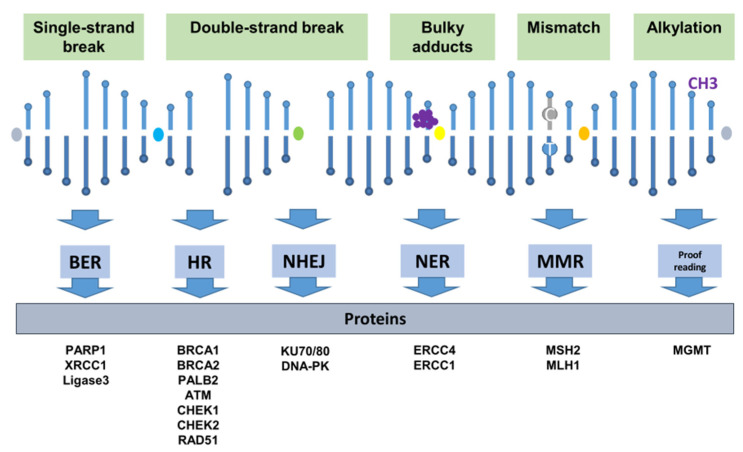
Major types of DNA damage, their according repair mechanisms and proteins involved. BER (Base Excision Repair), HR (Homologous Repair), NHEJ (Non-homologous end joining), NER (Nucleotide Excision Repair), MMR (Mismatch Repair) and proofreading.

**Figure 2 jpm-11-00612-f002:**
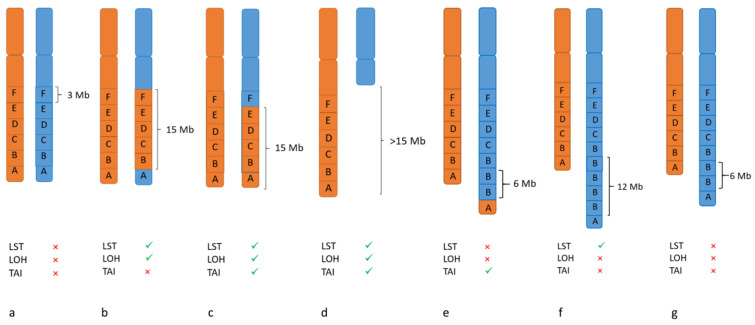
Examples for different alterations which lead to genomic scars. (**a**) Normal chromosomal pattern with maternal and paternal alleles (A–F); each box represents 3 Mb. (**b**–**g**) Alterations like rearrangements, loss or gain of chromosomal material that can result in positive scores for LOH (loss of heterozygosity), LST (large scale state transitions) and TAI (telomeric allelic imbalance).

**Figure 3 jpm-11-00612-f003:**
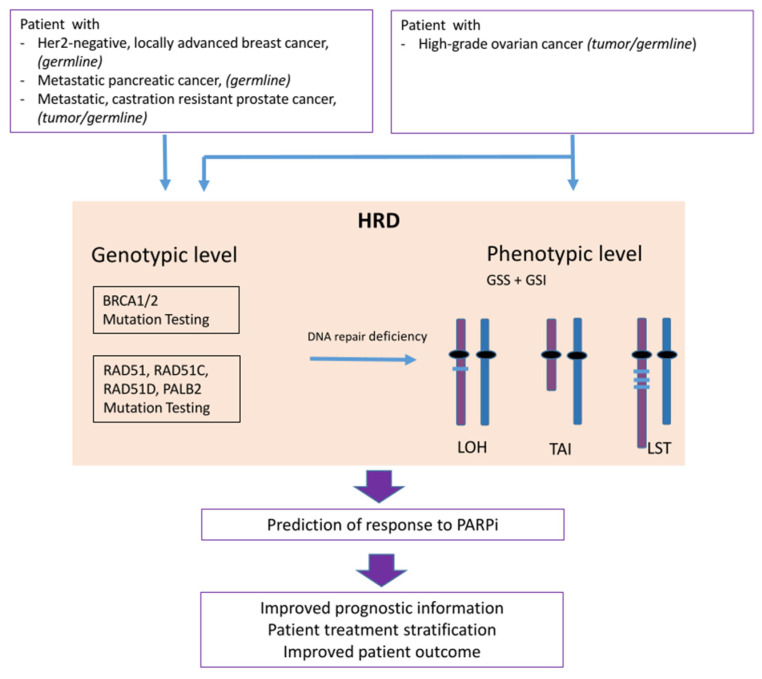
Flow chart and concept of homologous recombination deficiency (HRD) testing; from patient sample over genotyping and phenotyping to prediction of response to PARP inhibition (PARPi). Loss of heterozygosity (LOH), telomeric allelic imbalance (TAI) and large scale state transitions (LST), genomic instability score (GSI), genomic scare score (GSS).

**Table 1 jpm-11-00612-t001:** Different panel for the detection of Homologous recombination repair (HRR) genes and homologous recombination deficiency (HRD) including loss of heterozygosity (LOH), telomeric allelic imbalance (TAI) and large scale state transitions (LST). Definitions of LOH, TAI and LST can be found in Section 3.3.

Panel	Provider	Applicable in Lab	Focus	Number of Genes	LOH	TAI	LST
BROCA Cancer Risk Panel	UW Medical Center Laboratory Medicine—Genetics Lab	no	Hereditary cancer predisposition syndromes/Ovarian cancer/Breast cancer	74	no	no	no
BRCANext^TM^	AmbryGenetics	no	Hereditary Gynecological carcinomas and/or Breast cancer	18	no	no	no
BRCANext-*Expanded*™	AmbryGenetics	no	Hereditary Gynecological carcinomas and/or Breast cancer	23	no	no	no
CAN02	CeGaT	no	Ovarian cancer/Breast cancer	13	no	no	no
CAN21	CeGaT	no	Ovarian cancer/Breast cancer	41	no	no	no
Myriad myRisk^®^ Hereditary Cancer	Myriad	no	Hereditary cancer predisposition syndromes/Ovarian cancer/Breast cancer	36	no	no	no
Myriad’s myChoice^®^ CDx	Myriad	no	homologous recombination deficiency score/BRCA1,2	2	yes	yes	yes
Oncomine BRCA Expanded Panel	Thermo Fisher Scientific	Yes	Ovarian cancer/Breast cancer and Prostate carcinomas	15	no	no	no
Oncomine HRR Pathway Predesigned Panel	Thermo Fisher Scientific	Yes	Homologous recombination repair genes and other	28	no	no	no
Oncomine Comprehensive Plus	Thermo Fisher Scientific	Yes	Pan-Cancer	500+	yes	no	no
Trusight Tumor 170	Illumina	Yes	Pan-Cancer	170	no	no	no
TrueSight Oncology 500	Illumina	Yes	Pan-Cancer	500+	no	no	no
CentoBreast^®^	Centogene	no	Ovarian cancer/Breast cancer	30	no	no	no
CentoCancer^®^	Centogene	no	Pan-Cancer	70	no	no	no
HANDLE HRR NGS Panel	AmoyDx	yes	Homologous recombination repair genes and other	32	no	no	no
HRD Focus Panel	AmoyDx	yes	homologous recombination deficiency score/BRCA1,2	2	yes	yes	yes
Invitae Common Hereditary Cancers Panel	Invitae	Yes	Hereditary cancer predisposition syndromes/Ovarian cancer/Breast cancer	47			
FoundationOne^®^CD	Foundation Medicine	no	Pan-Cancer	324			
FoundationFocus CDxBRCA LOH	Foundation Medicine	no		2	yes	no	no
NeoTYPE^®^ HRD+ Profile	NeoGenomics	no	homologous recombination deficiency/Homologous recombination repair genes	30	no	no	no
QIAseq Homologous Recombination Repair (HRR) Panel	Qiagen	yes	Homologous recombination repair genes	15	no	no	no
SureMASTR HRR	Agilent	yes	Homologous recombination repair genes	17	no	no	no

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
