# Peer review of "Biomarkers for Homologous Recombination Deficiency in Cancer"

_jpm, 2021, doi:10.3390/jpm11070612_

Round 1

Reviewer 1 Report

In this review article, Svenja Wagener-Ryczek et al provided a comprehensive summary of biomarkers in HR deficiency caner treatment. In covers a very broad topics, from the fundamental principle to clinical/lab application, namely general DNA damage and DNA repair, HR deficiency, and PARPi (synthetic lethality concept, genes involved, available panel, and their detection methods and assessment). Obviously, this topic has been reviewed many times and many places (journals), mostly in the context of a specific cancer. The question here is how this one is better or different from others? Overall, this is a good review and will be interest of the readers at the Personalized Medicine. Here are some issues should be addressed:

  1. Overall, spelling, grammar and language should be checked thoroughly.
  2. Title: obviously the title is bigger than what this review is talking about.
  3. Abstract:

Homologous Recombination Deficiency (HRD), not the deficiency of…; define HRR and BRCAness. (whether simply name HRR as HR, and make it easy to compare HRD)

“the exact mechanism of”, unclear what is “the exact mechanism of”?

“Moreover the according biomarker assays”, unclear to me what is “the according…”

Abstract needs re-write. It is hard to follow the flow. For example, two “Nevertheless”, Nevertheless is used to connect two sentences that have the same topic, but I did not see here.

  1. Keywords, you mean “DNA double strand” or “DNA double-strand break”. I think PARPi should be included. Unclear to me why you pick up “parallel sequencing”?
  2. Line 47, add PARPi
  3. “1.1. The role of DNA repair in cancer” and “2. DNA repair mechanisms”. I think you talked about DNA repair mechanisms in section “1.1 The role of DNA repair in cancer”, but you did not talk about the connection of DNA repair and cancer in 1.1, which you should expand.
  4. 1, the cartoon of dsDNA is not clear to me. I got what you want to show in bulky adducts and mismatch, but what’s the difference among ssDNA break, dsDNA break, and alkylation or what you wanted to show? Figure legend, HR (homologous recombination). Spelling: repair (not reppair)

Proofreading, since you show it here you better explain it in text.

  1. Line 126, DNl4 or DN14?
  2. Line 140, Ref
  3. Table 1. Overall, Table 1 takes a lot of space, I am not sure how informative it is to readers? How about remove all “applicable in lab-No” panel, and comment on advantage and disadvantage on the left (applicable in lab-Yes).

The concept of LOH, TAI, and LST needs to be defined in text. Line 261-275 explain them, but not clear to me, especially TAI.

The number of genes, I know it is impossible to list 500+, but please list 2 genes.

What’s the rationale to put them in this order? The time when the panel was developed?

  1. 2, move it before Table 1 because LOH, TAI, and LST were listed in Table 1? Since 1 block =3 Mb, no need show 6 Mb for 2 blocks, 15 Mb for 5 blocks.
  2. Line 302, cromosomal should be chromosomal
  3. Line 315, chromosomal microarray analysis (CMA), lots of abbreviations have been used in this manuscript. To make it more readable, try to avoid create more abbreviations if they are just used 1 or 2 times.
  4. The concept of genomic scar score (GSS), genomic instability score (GIS), HRD score, LOH score are interesting and clinically useful. Whether authors can draw a flow chart: from patient sample, sequencing, BRCA1/2 mut or BRCAness, assessment by these scores, to PARPi, even to outcome? This is particularly attractive for readers like Personalized Medicine.
  5. Conclusion and Future Perspectives: more and detailed perspectives will be appreciated by readers.

Reviewer 2 Report

Defects in homologous recombination, while place a high-risk factor in cancer formation, render cancer cells hypersensitive to PARP inhibitors, which trap Poly(ADP-ribose) polymerase on DNA, block DNA replication, and trigger DNA double-strand break formation. Hence, PARP inhibitors become a promising therapeutic regime in the treatment of cancer patients with germline mutations that cause defects in homologous recombination. The review article by Wagener-Ryczek et.al,  overviewed the DNA repair pathways and the association of HR defect with treatments by PARP inhibitors. The authors further reviewed the biomarkers and the methods for their diagnosis, which serves as a critical step in the application of PARP inhibitors. The review is well compiled and will be of interest to the people in the field. I have a few comments on the background of the DNA repair pathways as depicted below that will further strengthen this review.

  • Page 3, lines 87-88. The removal of the damaged base by glycosylase results in an abasic site, which needs to be processed by AP endonuclease to create a DNA nick. The nick recruits PARP1 and is further repaired by Polbeta and the DNA ligase in the short-patch BER pathway.
  • Page 3, in Figure 1 legend, it should be “NHEJ”.
  • Page 3, the 2.1 and 2.2 titles may be better as “single-strand lesion repair” and “double-strand break repair”.
  • Page 4, lines 103-104, HR requires sister chromatid which exists following DNA replication. Hence, HR is preferred in S/G2 and NHEJ is dominant in the G1 phase, which is much longer compared to S/G2. This is likely an important reason for NHEJ being predominant in DSB repair in human cells.
  • Page 4, the paragraph of 2.2.1 needs revision. ATM is activated by MRN and ATR recruitment requires 5’ end resection. In Line 115-116, for resection, EXO1 continues after MRN and digests duplex DNA. DNA2 acts on 5’ single-stranded DNA and requires BLM helicase to unwind the duplex. “degrade the single-strand DNA synthesized in the first step” is incorrect. In line 120, “S2” should be “S”.
  • Page 4, Paragraph 2.2.2 reviewed the NHEJ pathway in yeast. The NHEJ pathway in humans should be reviewed instead, which involves the human-specific DNA-PK complex.  
  • Page 10, line 302, “chromosomal” should be “chromosomal”.

Round 2

Reviewer 2 Report

The authors have address my comments well.